# OpenReview forum: "DAMA: Data- and Model-aware Alignment of Multi-modal LLMs"
_ICML.cc/2025/Conference — ICML 2025 poster_

### Official Review · Reviewer_gFQs · 2025-03-11

**Overall Recommendation:** 4

**Summary:**

In this paper, the authors propose DAMO, an innovative data- and model-aware alignment strategy for Multi-modal Large Language Models (MLLMs), Specifically, a data-aware strategy is introduced to enhance the model's adaptability with data hardness, a model-aware is proposes to facilitate a more effective optimization with model's current responses. The authors conduct extensive experiments for evaluation, and the promising results demonstrate their effectiveness.

**Claims And Evidence:**

The authors' claims regarding improving Multi-modal Large Language Model (MLLM) alignment through data- and model-aware strategy are well-supported. Extensive experiments across multiple datasets with multiple metrics demonstrate consistent improvements compared to baseline methods. The results effectively demonstrate that enhancing MLLM alignment via the data- and model-aware strategy improves the alignment performances.

**Essential References Not Discussed:**

The paper adequately cites relevant works in multi-modal large language models and preference alignment.

**Experimental Designs Or Analyses:**

The experimental designs are comprehensive and valid. The authors evaluate their method on diverse benchmarks, demonstrating consistent performance improvements across different scenarios.

**Methods And Evaluation Criteria:**

The proposed method logically integrates data hardness and model responses by modulating the $\beta$, strengthens effectiveness withour introducing additional computations. The evaluation criteria for different benchmarks are standard and appropriate for MLLM tasks. The use of multiple benchmarks, along with comparisons to existing methods, validates the evaluation framework's robustness.

**Other Comments Or Suggestions:**

See weaknesses.

**Other Strengths And Weaknesses:**

Strengths:
1. Integrating data- and model-knowledge to MLLM alignment is an innovative idea, bringing new perspectives to the MLLM community.
2. The paper is well-organized, the figures to illustrate the problems and approaches are clearly.
3. Extensive evaluations, encompassing both quantitative metrics and qualitative assessments, demonstrate the effectiveness of the paper.

Weaknesses:
To deepen the understanding of the proposed approach, the authors could provide more detailed analysis to demonstrate how the data- and model-aware strategy influences performance, for instance, visualize and analyze the dynamics of $\beta$ during the training procedure for further illustration.

**Questions For Authors:**

See weaknesses.

**Relation To Broader Scientific Literature:**

This work makes significant contributions to the field of multi-modal large language models and preference alignment. The proposed method advances the state-of-the-art by demonstrating how data hardness and model responses can be effectively and efficiently integrated into MLLM alignment.

**Theoretical Claims:**

The theoretical foundation of the method is sound. The approach is based on the DPO method, which is well-supported by experimental results.

---

> ### Author Rebuttal · Authors · 2025-04-01
>
> Response to Reviewer $\color{green}\text{gFQs}$:
>
> We sincerely thank you for your invaluable and constructive feedback. We particularly appreciate your positive acknowledgement of our novelty, clear organization, and extensive experimental validations. Below we provide the point-to-point responses to address your concerns about our approach.
>
> > **In-depth Analysis of $\beta$.** To deepen the understanding of the proposed approach, the authors could provide more detailed analysis to demonstrate how the data- and model-aware strategy influences performance...
>
> Thank you for pointing this out. Your suggestion is invaluable for this paper. To address this, we show the dynamic $\beta$ concerning the data hardness, model responses, and the combination in the Figures 4,5,6 of **[the Anonymous Link](https://anonymous.4open.science/r/Rebuttal_DAMO-05C2/README.md)**.
>
> - **1. $\beta_{D}$ for the data hardness (Figure 4 in [Anonymous Link](https://anonymous.4open.science/r/Rebuttal_DAMO-05C2/README.md)).**
> From Figure 4, we can observe that the range of $\beta_{D}$ falls within (0.0524, 0.1428) with the original $\beta$ initialized as 0.1. Moreover, we observe that the mean value is 0.0999 with a standard deviation of 0.0288. These observations demonstrate that $\beta_{D}$ maintains proximity to the original $\beta$, while adaptively adjusting based on the data characteristics, enabling a more effective capture of the data.
>
> - **2. $\beta_{M}$ for the model responses (Figure 5 in [Anonymous Link](https://anonymous.4open.science/r/Rebuttal_DAMO-05C2/README.md)).**
> From Figure 5, we can find that as the training progresses, the $\beta_{M}$ gradually converges to the original $\beta$. Meanwhile, we find that the $\beta_{M}$fluctuates within a moderate range of (0.0530, 0.1580), demonstrating controlled adaptivity. These observations suggest that as the model training stabilizes, its responsiveness becomes more consistent and eventually approaches a steady state.
>
> - **3. $\beta_{C}$ by combining both (Figure 6 in [Anonymous Link](https://anonymous.4open.science/r/Rebuttal_DAMO-05C2/README.md)).**
> From Figure 6, we can observe that combining both the data- and model-aware strategies yields a more dynamic range of beta values, spanning from 0.0177 to 0.2261, which is wider than either. Moreover, while the value eventually stabilizes around 0.1, we notice that the mean value during the training stage is a bit lower than $\beta_{M}$. This suggests that it relaxes the constraints during training based on the data hardness, enabling the model to better capture fine-grained data patterns and thereby adaptively enhancing its responsiveness to data characteristics.
>
> Thank you again for your kind suggestions and supportive feedback. If you have any additional questions, we would be pleased to discuss them with you.

---

### Official Review · Reviewer_TUJo · 2025-03-13

**Overall Recommendation:** 4

**Summary:**

The paper examines the inherent property of DPO regarding its imbalanced responsiveness to data with varying difficulty levels and proposes Data and Model-aware DPO (DAMO) to address this issue. Experiments across various benchmarks demonstrate that DAMO enhances both trustworthiness and general task performance.

**update after rebuttal**

The authors have effectively addressed my principal concerns pertaining to the implementation details and have supplemented their work with additional experiments that underscore the generalizability of DAMO. Therefore, I have opted to retain my initial rating.

**Claims And Evidence:**

All pivotal claims in the paper are supported by empirical interpretation or systematic experimental validation.

**Essential References Not Discussed:**

Works directly relevant to contextualizing the paper’s contributions are appropriately cited and discussed.

**Experimental Designs Or Analyses:**

The ​performance validation in the article (e.g., the selection and analysis of ​hallucination benchmarks reflecting trustworthiness and ​general benchmarks) and the ​construction of ablation studies are ​methodologically sound and comprehensive.

**Methods And Evaluation Criteria:**

The proposed method DAMO is rational in tackling the imbalanced responsiveness issue, and the benchmark selection also makes sense.

**Other Comments Or Suggestions:**

Line 208 (right): "the momentum γ is set to 0.9, and **$\bar{H}$** is initialized to 0." **$\bar{H}$** should be **$\bar{R}$**.

**Other Strengths And Weaknesses:**

**Strengths:** The article demonstrates ​high writing quality with ​a well-organized content structure, ensuring ​ease of comprehension throughout.

**Weaknesses:** The paper ​lacks implementation details for the ​response split procedure in data-aware preference optimization. Specifically, the ​prompt templates used in LLaMA3 and ​examples of partitioned sub-sentences should be reported.

**Questions For Authors:**

1. Is the ​probability difference in ​Equation (6) sensitive to the ​LLM’s sub-sentence segmentation strategy, particularly regarding the ​number of sub-sentences generated?

2. If such sensitivity exists, how does the ​segmentation granularity empirically impact the ​performance of DAMO?

**Relation To Broader Scientific Literature:**

The paper's key contributions are primely related to Direct Preference Optimization (Rafailovetal., 2024).

**Theoretical Claims:**

All theoretical claims (the function and design of data-aware/model-aware preference optimization), including the proofs of key formulas, are accurate and validated.

---

> ### Author Rebuttal · Authors · 2025-04-01
>
> Response to Reviewer $\color{red}\text{TUJo}$:
>
> We highly appreciate your insightful comments and acknowledgment of our contributions! Your constructive criticism is invaluable in refining our work! We organize your concerns into the following 3 aspects:
>
> > **Q1. Clarification about sub-sentence construction.**
>
> Thank you for pointing this out! We provide our detailed prompt template as:
> ```
> You are an expert in extracting facts from the given question-answer pair about an image. Your task is to: Analyze the provided question-answer pair based on the image, extract all factual statements from the answer, and rewrite them into self-contained sentences.
> \n\n Requirements for each sentence are: \n1. complete, each sentence must be self-contained; \n2. factual (omit opinions, subjective statements); \n3. concise (no more than 77 tokens).
> \n\n Format your result strictly as:\n### Facts:\n- {Fact 1 (e.g., "A red shoe sits on a wooden floor.")}\n- {Fact 2 (e.g., "The shoe has laces and a white sole.")}\n- ...\n\n### Question-answer pair: Question: "{Question}" Answer: "{Answer}"
> ```
> For the very limited number of sub-sentences with more than 77 tokens (2/113156 for preferred, 1/114268 for rejected), we apply truncation over them.
> Here is a representative example demonstrating our sub-sentence construction:
> ```
> "question": "Is this book related to Literature & Fiction?",
> "answer": "No, this book is not related to Literature & Fiction. It is a religious or theological book, as evident from the title \"What Love Is This? Calvinism's Misrepresentation of God\" by Dave Hunt."
> "facts": [
> "The book is not related to Literature & Fiction.",
> "The book is a religious or theological book.",
> "The title of the book is \"What Love Is This? Calvinism's Misrepresentation of God\" by Dave Hunt."]
> ```
> > **Q2. Evaluation of different sub-sentence granularities.**
>
> Thank you for pointing this out! To validate this, we modify the prompt template by replacing the `77` in `no more than 77 tokens` with `60` and `50`.
>
> Table 1: Token Length of the segmented sub-sentences from the preferred responses (22626 responses in total)
>
> | Tokens | > 77 | 60-77 | 50-60 | 40-50 | 30-40 | 20-30 | < 20 | Total sub-sentences|
> |-|-|-|-|-|-|-|-|-|
> | `less than 77 tokens` | 0.002% | 0.005% | 0.047% | 0.540% | 5.161% | 30.659% | 63.586% | 113,156 |
> | `less than 60 tokens` | 0.007% | 0.004% | 0.048% | 0.536% | 5.176% | 30.778% | 63.451% | 113,040 |
> | `less than 50 tokens` | 0.006% | 0.004% | 0.040% | 0.532% | 5.194% | 30.734% | 63.490% | 113,085 |
>
> Table 2: Token Length of the segmented  sub-sentences from the rejected responses (22626 responses in total)
> | Tokens | > 77 | 60-77 | 50-60 | 40-50 | 30-40 | 20-30 | < 20 | Total sub-sentences|
> |-|-|-|-|-|-|-|-|-|
> | `less than 77 tokens` | 0.001% | 0.014% | 0.066% | 0.554% | 5.228% | 30.630% | 63.507% | 114,268 |
> | `less than 60 tokens` | 0.001% | 0.010% | 0.067% | 0.562% | 5.264% | 30.552% | 63.544% | 114,247 |
> | `less than 50 tokens` | 0.006% | 0.004% | 0.048% | 0.524% | 5.194% | 30.734% | 63.490% | 114,218 |
>
> Tables 1 and 2 show the statistics of the segmented sub-sentences. The statistics strongly support the robustness of our approach, with over 99% of sub-sentences containing fewer than 40 tokens, and more than 63% having fewer than 20 tokens.
>
> Due to the inherent inability of the LLM to precisely handle the length constraints [1], we manually truncate the sub-sentences exceeding the length constraints at different scales. The distributions of $\delta$ are in Figures 1,2,3 of **[the Anonymous link](https://anonymous.4open.science/r/Rebuttal_DAMO-05C2/README.md)**, and subtle differences in $\delta$ can be observed over different segmentation strategies.
>
> Table 3: Performance over object-hall bench.
>
>  method | response | mention
> -|-|-
>  `less than 77 tokens` | 82.54 | 90.64
>  `less than 60 tokens` | 82.25 | 90.08
>  `less than 50 tokens` | 81.78 | 90.20
>
> Moreover, we also evaluate the model trained with the given different $\delta$ in Table 3, and observe subtle differences. These comprehensive statistics and experimental results demonstrate DAMO's robustness across different segmentation granularities.
>
>
> > **Q3. Typo Correction.**  Line 208 (right): "the momentum $\gamma$ is set to 0.9, and $\bar{H}$ is initialized to 0." $\bar{H}$ should be $\bar{R}$.
>
> We acknowledge this typographical error and have corrected this in the revised paper.
>
> Thank you again for your valuable and insightful suggestions. We welcome any additional questions and would be happy to discuss them further.
>
> [1] Yuan et al. "Following length constraints in instructions."

---

### Official Review · Reviewer_ERND · 2025-03-21

**Overall Recommendation:** 4

**Summary:**

Authors propose a variant of DPO where the Beta hyperparameter is adapter dynamically depending on model and data- awareness. Author postulate the existence of easy and hard to distinguish example in alignment training settings, and therefore propose dynamic strategy to adjust those. Evaluation is reported on 5 benchmarks, including Object HalBench.

**Claims And Evidence:**

- Authors claim that introducing more regularization through Beta for easy to distinguish examples, and more regularization at the batch level help the model to learn from preferences, leading to improved results on Hallucinations benchmarks. However, this claim might be problematic to verify because all the experiments have been conducted on LLaVA 1.5, which is model lagging fairly behind on current VLM benchmarks compared to more modern alternatives. One could have probably found this work more compelling if those experiments would have been conducted on a more modern LLaVA (e.g LLaVA 1.6) or different models (e.g. InternVL2.5 or QwenVL2).
- Besides, this work only focuses on hallucinations, while there is clearly a link with the helpfulness of the responses given by aligned models. A non-helpful model with short answers will always lead to less hallucinations, and hence improved scores on the reported Benchmarks. The claims made in this work would be better supported with an extended view of the problem, ie. not just the hallucinations.
- Finally, while the formalization proposed in this work is extensive, and at time even a bit scholar like in Section 2, one could have preferred having a more extended experimental setup with more than just LLaVA 1.5 and more than one training dataset.

**Essential References Not Discussed:**

- Understanding Alignment in Multimodal LLMs: A Comprehensive Study, Amirloo et al., 2024, performed alignment evaluation on LLaVA 1.6. This is unclear why the work evaluated here (a) report experiment in LLaVA 1.5 instead of LLaVA 1.6, and (b) does not compare results with Amirloo et al. 2024 on that very similar topic.

**Experimental Designs Or Analyses:**

- The LLaVA 1.5 model has been used in numerous multimodal alignment papers in 2024. This is surprising to see it explored in this work yet another time, with a training dataset that was already introduced in the RLAIF-v paper. There is a possible experimental flaw in constraining this work in that small experimental setting.

**Methods And Evaluation Criteria:**

- Authors focus on hallucinations, which is a common pattern in recent Multimodal Alignment papers, but seem to ignore Helpfulness altogether, where the 'preferred response' is more helpful than the 'rejected response'. However, one could argue there is a clear relationship between helpfulness and hallucinations: it's easy to trick the hallucinations benchmarks reported in this work with shorter answers where the likelihood to produce hallucinations is inherently smaller but so the helpfulness. One could have appreciated benchmarks beyond hallucinations, along report on the responses length.
- In that regard, it is not surprising to see no improvement on the LLaVA-Bench benchmark, which arguably might be the only benchmark that might give a measure of the helpfulness of a model.

**Other Comments Or Suggestions:**

- Please do not mix serif (body-text, Figure 4) and non-serif font (Figure 1, 2, 3).
- L081: "Similarly to Section 3.1". It is odd to see this statement in Section 1. Is that intended?
- Generally the Paragraph L079-L086 does not explain what is Model-aware Preference Optimization. We learn it is similar to another section, but it's not stated how exactly the gap between chosen vs. rejected is used to scale Beta. Can you try to (a) make the overall process explicit, and (b) avoid referring future sections when the structure of the paper is not yet introduced?
- All the Section 2 could be removed from the paper has that topic has been already presented in many other papers? For instance (Tang et al., 2024 - Generalized Preference Optimization: A Unified Approach to Offline Alignment) or (Tang et al., 2024 - Understanding the performance gap between online and offline alignment algorithms) are clear reference in that matter. In particular, it is a bit scholar to introduce PPO when this work is about offline direct alignment.
- Equation 8 is there an extra dot after D?

**Other Strengths And Weaknesses:**

- Two important weaknesses of this work are (a) the lack of experimental novelty, - this work uses LLaVA 1.5 like numerous other papers before, and train it on the RLAIF-V preferences, exactly like Yu et al, 2024, (b) the single dimension view of the problem, where the focus is on hallucinations, - leaving alone the helpfulness of the model. It is likely that the aligned model might be less helpful and provide vague answers, explaining the higher scores on hallucinations benchmark, but the lack of improvement on LlaVA-Bench.

**Questions For Authors:**

Dear authors,
- Have you considered evaluating the helpfulness of your model along the hallucinations? Optimizing only for hallucinations might introduce an important bias in the helpfulness of the responses of the aligned model. One can easily trick Hallucinations benchmark with overly short or vague answers. The lack of improvement on Llava-Bench, arguably the only benchmark evaluating the helpfulness, tend to show that direction.
- Have you considered using another model than LLaVA 1.5, which by today standard is fairly old and has been already explored by 10+ other multimodal alignment papers since 2023?

**Relation To Broader Scientific Literature:**

Numerous papers have been published in 2023 and 2024 about using LLaVA 1.5 for alignment. Today, in 2025, one hopes to see the community explore further experimental settings beyond LLaVA 1.5. This work here explores the pair LLaVA 1.5 + RLAIF-V preferences that have been already explored in a previous work. The novelty lies only in the defining of the regularization parameter B of the DPO loss.

**Theoretical Claims:**

- See 'Claims And Evidence'.

---

> ### Author Rebuttal · Authors · 2025-04-01
>
> Response to Reviewer $\color{blue}\text{ERND}$:
>
> We highly appreciate your insightful comments, which help us a lot to better scrutiny and polish our work! The following are point-to-point responses.
>
> > **Q1. Implementation with more advanced models (e.g., LLaVA 1.6 and LLaVA-OneVision) makes DAMO more compelling.**
>
> Thank you for your kind suggestion. We extend DAMO to more advanced models like LLaVA 1.6 and LLaVA-one-vision with a more comprehensive alignment dataset[1]. Due to the GPU constraint, **only the MLP layer is finetuned for LLaVA-one-vision**, and all models are finetuned with one epoch. As shown in Table 1, DAMO still demonstrates consistent performance gains across these advanced models.
>
> Table 1: Performance on the Object-hal bench.
>
> Model | Response | Sentence | Average Length
> --|--|--|--
> LLaVA-1.6 | 84.36 | 91.10 | 194.75
> LLaVA-1.6 + DAMO | **85.61** | **92.39** | 182.06
> LLaVA-OneVision | 81.48 | 90.29 | 223.08
> LLaVA-OneVision + DAMO | **85.82** | **91.22** | 243.86
>
>
> > **Q2. Measurements over more helpfulness benchmarks further support our effectiveness.**
>
> Thank you for your kind suggestion regarding model helpfulness. Our evaluation on **LLaVA-Bench and MM-Vet** (Table 4 of the paper) shows that both our 7B and 13B models achieve competitive performance, with notable improvements of 6% and 9% over their baseline counterparts, respectively.
>
> Furthermore, we extend the helpfulness evaluation to the MME Bench with advanced models. As shown in Table 2, DAMO consistently improves both perception and cognition capabilities across different model architectures, demonstrating its effectiveness in enhancing both helpfulness and hallucination resistance.
>
> Table 2: Performance on the MME bench.
>
> Model | Perception | Cognition
> --|--|--
> LLaVA-1.6 | 1498 | 286
> LLaVA-1.6 + DAMO | **1503** | **291**
> LLaVA-OneVision | 1565 | 335
> LLaVA-OneVision + DAMO | **1571** | **339**
>
> **Notes:** We sincerely appreciate your valuable feedback regarding the helpfulness and advanced models. Our extensive experiments demonstrate that DAMO achieves consistent and significant improvements across:
>
> 1. Multiple model architectures (LLaVA-1.5 / 1.6, LLaVA-one-vision)
> 2. Extensive benchmark suites, including:
> 	- Helpfulness (MME, MMVet, LLaVA-Bench)
> 	- Hallucination (Object-Hal, AMBER, MM-Hal)
>
> **Most importantly**, DAMO serves as a plug-and-play mechanism that can be seamlessly integrated into various architectures with different alignment data, while maintaining minimal computational overhead. This versatility and efficiency, combined with consistent performance gains across different settings, support the effectiveness and practical value of DAMO.
>
> > **Q3. Adding essential related work.** Understanding Alignment in Multimodal LLMs: A Comprehensive Study, Amirloo et al., 2024.
>
> Thank you for providing this related work. We have added this method to our revised paper, `As a representative method, BDHS significantly advances the MLLM community by pioneering the alignment techniques into the advanced LLaVA 1.6 architecture, effectively bridging the gap between theoretical MLLM alignment research and practical applications.`
>
> > **Q4. Clarification about the model-aware strategy.**
>
> Sorry about the confusion. Let us clarify our model-aware strategy that adjusts $\beta$ according to the implicit reward gap between the preferred $y_w$ and rejected $y_l$. Specifically:
> 	1. A larger $\beta$ is assigned to a larger reward gap between $y_w$ and $y_l$, which indicates that the model has already grasped this type of response well.
> 	2. A smaller $\beta$ is assigned to a smaller reward gap between $y_w$ and $y_l$, which suggests that the model needs to improve its responsiveness on such cases.
> This adaptive scaling mechanism helps the model focus more on cases with less confidence, while maintaining its performance on well-learned cases.
>
> > **Q5. Presentation refinement.**
>
> * Please do not mix serif (body-text, Figure 4) and non-serif font (Figure 1, 2, 3).
>
> Thank you for pointing this out. We have unified the font in our revised paper.
>
> * L081: "Similarly to Section 3.1". It is odd to see this statement in Section 1.
>
> Thank you for your kind suggestions. We have removed such descriptions and polished this as discussed in **Q4**.
>
> * All Section 2 could be removed ...
>
> While this section provides essential background for researchers new to preference alignment in MLLMs, we have significantly condensed it to improve paper conciseness.
>
> * Equation 8, is there an extra dot after D?
>
> Sorry about the confusion. The first dot is the dot product, and the second dot is the period at the sentence end. We have modified as $\beta_{D} = \beta \times \alpha_{D}$.
>
> * one could appreciate to have a panel of responses in the appendix ...
>
> Thank you for constructive suggestions, we have added these analyses to the revised paper.
>
> [1] Yi-Fan Zhang etal MM-RLHF: The Next Step Forward in Multimodal LLM Alignment, 2025.

---

### Decision · Program_Chairs · 2025-05-01

**Decision:**

Accept (poster)

**Comment:**

This paper introduces DAMO, a data- and model-aware enhancement to align multi-modal LLMs. The method adaptively tunes alignment strength based on data difficulty and model confidence, addressing a key gap in prior work. While initial concerns of reviewers focused on limited model and benchmark diversity, the authors provided substantial new experiments on LLaVA 1.6 and OneVision, alongside helpfulness metrics and fine-grained beta analysis, convincingly addressing all major criticisms. I recommend acceptance.